Prevalence, patterns, and determinants of drug-resistant tuberculosis in Gulf Cooperation Council countries: an updated systematic review

Alibrahim Alaa 1 Aoalbrahim@ju.edu.sa
Alqahtani Homoud 2
Thirunavukkarasu Ashokkumar 3
Qazi Ibtisam 3
1 Department of Internal Medicine, College of Medicine, Jouf University , Sakaka, Aljouf , Saudi Arabia
2 Department of Public Health, Hafr Al Batin Health Cluster , Hafr Al Batin , Saudi Arabia
3 Department of Family and Community Medicine, College of Medicine, Jouf University , Sakaka, Aljouf , Saudi Arabia
Krajaejun Theerapong
Electronic publication date: 2024 Nov 28
Publication date: 2024
Volume: 12
Electronic Location ID: e18628
Received 2024 Jul 8; Accepted 2024 Nov 12
Copyright: © 2024 Alibrahim et al.
Copyright year: 2024
Copyright holder: Alibrahim et al.
License: This is an open access article distributed under the terms of the Creative Commons Attribution License, which permits unrestricted use, distribution, reproduction and adaptation in any medium and for any purpose provided that it is properly attributed. For attribution, the original author(s), title, publication source (PeerJ) and either DOI or URL of the article must be cited.
License URL: https://creativecommons.org/licenses/by/4.0/

Keywords: Drug resistance, Tuberculosis, Isoniazid, Multidrug, Previous treatment, GCC

Funding: Deanship of Graduate Studies and Scientific Research at Jouf University DGSSR-2024-01-01052 This work was funded by the Deanship of Graduate Studies and Scientific Research at Jouf University under grant No. (DGSSR-2024-01-01052). The funders had no role in study design, data collection and analysis, decision to publish, or preparation of the manuscript.

==============================
Drug resistance (DR) to antituberculosis drugs is a growing global problem that threatens the successful control of tuberculosis (TB) globally and within the Gulf Cooperation Council (GCC). In the GCC, TB remains a major public health issue. Understanding the prevalence and patterns of drug resistance to antituberculosis drugs is crucial for developing effective prevention and treatment strategies. Hence, the present systematic review is aimed at assessing the prevalence, pattern, and risk factors of drug-resistant TB (DR-TB) in GCC countries. We conducted this systematic review adhering to the guidelines outlined in the Preferred Reporting Items for Systematic Review and Meta-Analysis (PRISMA) 2020 Statement. Using the relevant keywords in the major databases, we included peer-reviewed articles that were published from 01 January 2014 and onwards in English language journals. The prevalence and patterns of DR-TB levels in different countries were different. Isoniazid monoresistance was the most commonly found type of resistance, with varying degrees of prevalence of multidrug-resistant tuberculosis (MDR-TB). Risk factors for DR-TB included diabetes mellitus, past TB treatment, younger age, female gender, and renal failure. There was a positive correlation between expatriate status and DR-TB. Collaborative actions by relevant stakeholders are essential to implement evidence-based interventions that reduce the DR-TB burden and improve overall community health. Ongoing research and surveillance activities are necessary for monitoring patterns, identifying new risk factors, and providing focused interventions to lessen the threat of DR-TB on public health in GCC countries.

Introduction

Drug resistance (DR) to antituberculosis drugs is a growing global problem that threatens the successful control of tuberculosis (TB) globally. Despite declining trends in TB infection rates, death, and prevalence globally (Africa excluded), current research shows that TB remains among the top killer diseases worldwide. A total of 1.3 million people died from TB in 2022 (Bloom et al., 2017; World Health Organization (WHO), 2023a). The World Health Organization (WHO) has stated that about 18% of all TB cases in the world had shown multidrug resistant (MDR-TB): either to the first or second-line antituberculosis drugs, with another 5% resistant to a combination of rifampicin and isoniazid, which are stronger antituberculosis usually used in first-line treatment and 8.5% of the MDR-TB infection being extensively drug-resistant (XDR-TB) (World Health Organization (WHO), 2022a). Studies from the Kingdom of Saudi Arabia (KSA) show that the MDR-TB prevalence is approximately 5% among Saudi TB patients, with 17% being DR cases (Sambas et al., 2020; Al Ammari et al., 2018). In KSA, DR to anti-tuberculosis medications is especially concerning, as the country has one of the highest TB burdens in the world (Al Ammari et al., 2018; Elhassan et al., 2017; Asaad & Alqahtani, 2012). Research studies show that TB is a continuously growing health concern in many Middle Eastern and North African countries, with GCC countries being part of it (Asaad & Alqahtani, 2012; Barry, 2021). Various states and governments across the continents, in response, are employing various control and mitigation strategies and high standards of prevention as a way to tackle this health issue (World Health Organization (WHO), 2023b; Essack et al., 2017).

DR refers to a microorganism’s ability to resist antibiotic’s effects. “MDR-TB” is caused by an organism that is resistant to at least isoniazid and rifampin, the two most potent TB drugs (World Health Organization (WHO), 2022a). Drug-resistant tuberculosis (DR-TB) makes it harder for doctors to treat infections with antituberculosis drugs, and it can make some infections resistant to treatment. According to the CDC, DR-TB or MDR-TB resistance occurs when bacteria develop mechanisms to evade or overcome the action of antibiotics, like limiting drug intake, making them less effective against infections (Peterson & Kaur, 2018; Centers for Disease Control and Prevention (CDC), 2019). As a result, this can lead to serious health problems for people who are treated with antibiotics. Antituberculosis DR is a global problem that is getting worse yearly (World Health Organization (WHO), 2022a; Aslam et al., 2018). According to the WHO, antibiotic resistance is a major contributor to the development of MDR-TB, which is now a global threat (World Health Organization (WHO), 2022a). DR-TB is caused by several factors, including overuse of antibiotics, mutation of bacteria genes that make them resistant to antibiotics, and ineffective or outdated antibiotics (Aslam et al., 2018; Ventola, 2015). The WHO estimates that antibiotic resistance is responsible for at least two million deaths yearly, a growing global health threat (World Health Organization (WHO), 2022a).

In GCC countries, TB remains a major public health issue. Understanding the prevalence and patterns of drug resistance to antituberculosis drugs is crucial for developing effective prevention and treatment strategies in GCC countries where it remains a major public health issue (Saati et al., 2021; Al-Hayani et al., 2021). Continuous assessment of the magnitude of DR-TB and associated factors of DR to antituberculosis drugs is important as it can help us to understand the problem and devise ways to address this major public health issue. Moreover, conducting a new systematic review on DR-TB in GCC countries is warranted to provide updated, context-specific, and methodologically rigorous evidence to inform policy, practice, and future research in this critical area of public health. Hence, the present systematic review is aimed to assess the prevalence, patterns, and risk factors of DR-TB in GCC countries.

Materials and Methods

Registration

The present systematic review is registered with the International Prospective Register of Systematic Reviews (PROSPERO 2024-CRD42024513293).

PRISMA 2020 statement

We conducted this systematic review adhering to the guidelines outlined in the Preferred Reporting Items for Systematic Review and Meta-Analysis (PRISMA) 2020 Statement (Please find the PRISMA checklist in the Supplemental File—Appendix 1) (Page et al., 2021).

Eligibility criteria

We included all peer-reviewed articles that were published from 01 January 2014 and onwards in English language journals. The research team incorporated the studies that were conducted in GCC countries (Bahrain, Kuwait, Oman, Qatar, KSA, and the United Arab Emirates (UAE)), having participants diagnosed with TB, including both pulmonary and extrapulmonary TB cases, studies reporting data on DR-TB, including resistance to first-line drugs (isoniazid, rifampicin, ethambutol, pyrazinamide) and second-line drugs (fluoroquinolones, injectable agents, etc.). The research team excluded studies that solely evaluated drug-sensitive TB cases, outcomes unrelated to TB, animal studies, laboratory experiments, and in vitro studies. The above-mentioned inclusion and exclusion criteria applied in our study assisted us in refining the search strategies. They ensured that the present systematic review explicitly evaluated related to DR-TB within the GCC region, providing a clear and comprehensive analysis of the prevalence, patterns, and determinants of DR in TB cases in this region.

Information sources and search strategies

This systematic review utilized Medline, Saudi Digital Library (SDL), Google Scholar, and Web of Science (WoS) databases to find the articles relevant to our study. Keywords used in combination with Boolean operators (“AND” and “OR”) were the following: ‘Tuberculosis,’ ‘Drug-resistant tuberculosis,’ ‘Multidrug-resistant tuberculosis (MDR-TB),’ ‘Extensively drug-resistant tuberculosis (XDR-TB),’ ‘Prevalence,’ ‘Pattern,’ ‘Risk factors,’ ‘Determinants,’ ‘Treatment,’ ‘Epidemiology,’ ‘Diagnosis’. Furthermore, in keywords, we included ‘GCC countries’ or specified the name of the country that belongs to this region. At the beginning of the study’s selection steps, two researchers (Alaa Oqalaa E Alibrahim and Homoud Alqahtani) assessed each article’s eligibility. In the event of differences of opinion or disagreement between the assessments of these two researchers regarding the suitability of an article, a third researcher, Ashokkumar Thirunavukkarasu, who also serves as an author and the referee, was consulted. Ashokkumar Thirunavukkarasu, with expertise in the field and experience as an adult infectious disease specialist, facilitated focused group discussions where Alaa Oqalaa E Alibrahim and Homoud Alqahtani presented their findings, viewpoints, and evidence. These discussions typically lasted for an hour and followed a structured format to ensure all perspectives were thoroughly considered. Ashokkumar Thirunavukkarasu ensured that each author had an equal opportunity to present their arguments and provided decisive input when necessary. The final decision was reached through this collaborative process. All disagreements and decisions were documented and integrated into the study to maintain transparency and ensure the credibility and reliability of the included articles.

Data extraction

We extracted data from the selected articles and entered it into a Microsoft Excel sheet for data management and further assessment. Two more faculties of the present review double-checked the data curation process to minimize the error. Their role is restricted only to the data curation process. The data collation included the study’s characteristics (author et al., country, year of publication, study design), study participants’ characteristics (age, gender, etc.), drug resistance prevalence, patterns (mono, MDR-TB, XDR-TB), and determinants (risk factors).

Study risk of bias assessment

To evaluate the methodological quality and risk of bias of the articles included in the systematic review, the Joanna Briggs Institute (JBI) Critical Appraisal Checklists were utilized. It is vital to realize that no research is without flaws. Every study has certain limitations. The JBI Critical Appraisal Checklists, on the other hand, are widely used to identify studies with excellent methodological quality and minimal risk of bias. The JBI Critical Appraisal Checklists give a series of questions that evaluate the following characteristics of a study, including study layout, sample, data gathering, data examination, findings interpretation, and potential conflicts of interest. Using the JBI Critical Appraisal Checklist, the studies’ methodological quality was graded as high, moderate, or poor, and their risk of bias was rated as low (all questions responses were yes), moderate (one or two of the responses were no or unclear), or high (more than two questions responses were no or unclear). We included the studies that had a low and medium risk of bias. Quality assessment using the JBI critical appraisal checklist of included articles is presented in Table 1.

Table 1 Quality assessment of included studies using the JBI critical appraisal checklist.

Study	Inclusion criteria	Study subject and settings	Exposure measurement	Objectives/standard criteria	Confounders identified	Confounders control	Outcome measurement	Statistical tests	
Al Mahrouqi et al. (2022)	*	*	*	*	*	*	*	*	
Gaifer, Babiker & Rizavi (2017)	*	*	*	*	*	*	*	*	
Hegazy et al. (2021) **	*	*	*	*	–	–	*	*	
Al Ammari et al. (2018)	*	*	*	*	*	*	*	*	
Sambas et al. (2020)	*	*	*	*	*	*	*	*	
Al-Shahrani et al. (2021)	*	*	*	*	*	*	*	*	
Al-Hayani et al. (2021)	*	*	*	*	*	*	*	*	
Elhassan et al. (2017) **	*	*	*	*	–	–	*	*	
Binkhamis et al. (2021)	*	*	*	*	*	*	*	*	
Varghese & Al-Hajoj (2017)	*	*	*	*	*	*	*	*	
Ali et al. (2020)	*	*	*	*	*	*	*	*	
Khawaja, Jawad & Ghanem (2020) **	*	*	*	*	–	–	*	*	
Al-Mutairi et al. (2022)	*	*	*	*	*	*	*	*	
Habous et al. (2020)	*	*	*	*	*	*	*	*	
Notes:

* Each study fulfilled the criteria for this item in the JBI critical appraisal checklist.

** Medium risk of bias studies. Remaining studies had low risk of bias.

–The study did not fulfill the criteria for this item in the JBI critical appraisal checklist.

Results

Study selection process

The study selection process followed in the present systematic review is depicted in Fig. 1 (PRISMA Flowchart).

Figure 1 PRISMA diagram—study selection process.

The present systematic review’s results summary is presented in Table 2.

Table 2 Prevalence, pattern, and risk factors of DR-TB (including attributes of the included studies).

Study	Country	Sample size	Prevalence and pattern	Risk factors	
Al Mahrouqi et al. (2022)	Oman	2,192	341 (13.4%) M. tuberculosis strains showed resistance to any drug	The pattern of TB cases suggests transmission from the migrant population	
Of the 341, Mono-resistance (MR)—71%	
Poly-resistance (PR)—11.7%	
MDRTB—17.3%.	
Gaifer, Babiker & Rizavi (2017)	250	MDRTB Prevalence—1.8%.	Previous treatment for TB	
TB isolates resistant to any of the first-line TB drugs—7.5% of cases.	Female gender	
Commonest MR drug was Pyrazinamide—3.5%.	Younger age	
Hegazy et al. (2021)	115	Resistance to pyrazinamide was detected in six cases.	Not evaluated	
Rifampicin in three cases.	
Isoniazid, streptomycin, and kanamycin in one case each	
Al Ammari et al. (2018)	Saudi Arabia	2,098	4.4% of TB cases were found to have MDR-TB.	Younger age group, female gender, and those with a previous history of TB.	
MR Pattern: 3.8% for ethambutol, 5.4% for pyrazinamide, 10.2% for isoniazid, 11% for streptomycin, and 5.9% for rifampicin.	Renal failure—Risk factor for rifampicin resistance.	
Sambas et al. (2020)	158	Overall prevalence of DR—17.1% MDRTB—5%	Smoking, Age, previous history of TB and preexisting lung disease	
Among the resistant cases, streptomycin (25.9%) and isoniazid (11.1%) were the drugs most affected by resistance.	
Al-Shahrani et al. (2021)	901	Overall TB-DR—21.4%	Younger age	
Pattern:	
MR—91.7%	
MDRTB—8.3%	
Pyrazinamid/e had the highest prevalence MR—33.4%.	
Ethambutol had the lowest resistance—7.1%.	
Al-Hayani et al. (2021)	472	MDRTB—1.5%	Gender (Male)	
MR ranged from 2.1% (Pyrazinamide) to 3.4% (isoniazid and streptomycin)	
Elhassan et al. (2017)	622	MDRTB—4%	Not evaluated	
MR Pattern:	
Isoniazid—1.8%, rifampin—1.4%, streptomycin—1.9%, ethambutol—1.1%, and pyrazinamidev—2.1%.	
Binkhamis et al. (2021)	105	Isoniazid-MR tuberculosis—8.6%	The study did not find any significance in the outcome of isoniazid-resistant cases compared to non-isoniazid-resistant cases.	
Varghese & Al-Hajoj (2017)	2,956	MDR-TB—83 isolates	Higher prevalence of DR to second-line drugs among expatriates	
Moxifloxacin—10.8%	
Ofloxacin—7.2%	
Capreomycin—3.6%	
Amikacin—2.4%.	
Ali et al. (2020)	Qatar	3,301	6.7% resistance to at least one drug	Migrant population (Expatriates)	
MDRTB—1.2%	
Khawaja, Jawad & Ghanem (2020)	Bahrain	588	MDRTB—3%	Not evaluated	
MR pattern:	
Isoniazid—9%	
Streptomycin—6%	
Rifampicin—3%	
Ethambutol—1%	
Al-Mutairi et al. (2022)	Kuwait	256	15 isolates showed resistant to one or more antituberculosis drugs	Expatriate patients were infected with unique TB strains likely acquired in their native countries before arriving in Kuwait.	
Habous et al. (2020)	UAE	1,116	Overall TB-DR—17.3%	Diabetes mellitus	
MDRTB—17.3%	
MR pattern:	
Isoniazid—3.6%	
Streptomycin—2.9%	
Pyrazinamide—2.6%	
Rifampicin—0.3%	
Regarding DR pattern to second-line drugs among MDR-TB patients:	
Ofloxacin—2%	
Ofloxacin + Moxifloxacin—10%	
Amikacin + Capreomycin + Ofloxacin + Moxifloxacin—2%	

The PRISMA flowchart details the systematic review process of identifying and selecting studies on drug-resistant tuberculosis in GCC countries. Initially, a comprehensive search yielded 2,941 articles from the Saudi Digital Library (977), Google Scholar (1,041), and Web of Science (923). Following a screening of titles and abstracts, 2,683 articles were excluded for irrelevance. This left 258 studies for full-text review. Of these, 245 were excluded for reasons such as not being original research, not focusing on GCC countries, being over 10 years old, involving animal experiments, having irrelevant populations, or having a high risk of bias (as assessed by the JBI Checklist). Ultimately, 14 studies met the inclusion criteria and were incorporated into the systematic review.

Prevalence and patterns of DR-TB

The prevalence of MDRTB varied significantly across the countries studied. In Qatar, the prevalence was relatively low at 1.2%, with isoniazid monoresistance being the most common pattern observed (Ali et al., 2020). The distribution of resistance patterns varied, with isoniazid monoresistance consistently identified as the predominant pattern. In Bahrain, a study reported a relatively low prevalence of MDR-TB at 3% among TB patients, with isoniazid monoresistance being the most common pattern observed (Khawaja, Jawad & Ghanem, 2020). Similarly, in KSA, the prevalence of MDR-TB was 4.4%, with isoniazid monoresistance noted as the predominant resistance pattern. Additionally, rifampicin resistance was identified in 5.9% of TB patients in KSA (Al Ammari et al., 2018). In Oman, a higher prevalence of DR-TB affects 13.4% of TB patients, with a diverse range of resistance patterns observed, including mono-resistance, poly-resistance, and MDR-TB. Isoniazid monoresistance was the most common, followed by streptomycin and rifampicin resistance (Al Mahrouqi et al., 2022). Some authors explored the DR pattern to second-line drugs, including fluoroquinolones and second-line injectable drugs. Of the fifty MDR-TB patients, Habous et al. (2020) reported that 2% were resistant to Ofloxacin, 10% were resistant to Ofloxacin and Moxifloxacin, and 2% were resistant to four drugs (Amikacin, Capreomycin, Ofloxacin, and Moxifloxacin) (Habous et al., 2020) and Hegazy et al. (2021) found that one TB patient out of 115 had resistance to Kanamycin. Varghese & Al-Hajoj (2017) explored DR patterns and found 83 patients had MDR-TB. Among the MDR-TB isolates, they observed the highest proportion of resistance with moxifloxacin (about 11%), followed by other second-line drugs. In their study, the lowest level of resistance among MDR-TB isolates was observed in Amikacin (2.4%) (Varghese & Al-Hajoj, 2017).

Determinants of DR-TB

In UAE, a retrospective analysis revealed that diabetes mellitus (DM) was identified as a significant risk factor for DR-TB, with TB patients with DM showing higher rates of drug resistance compared to those without DM (Habous et al., 2020). The study reported that among TB-DM cases, the rates of DR-TB, including monoresistance, polyresistance, and multidrug resistance, were higher compared to TB patients without DM, with statistically significant differences being observed (p = 0.039). Additionally, previous TB treatment emerged as a significant risk factor for DR-TB in Oman and KSA, with a higher prevalence of drug resistance observed among patients with a history of previous TB treatment (p = 0.001, odds ratio (OR) = 14.81; 95% confidence interval (CI) [3.09−70.98]) (Al Ammari et al., 2018; Gaifer, Babiker & Rizavi, 2017).

Some studies in KSA identified several significant risk factors for drug-resistant TB, including age (p = 0.017, OR = 1.09, 95% CI [1.02–1.18]), gender (p = 0.015, OR = 2.21, 95% CI [1.17–4.18]), and previous history of TB treatment (p = 0.001, OR = 7.33–19.93). Renal failure was found to be a significant risk factor for developing resistance to rifampicin, with a statistically significant association (p = 0.022, OR = 6.61, 95% CI [1.86–23.50]) (Sambas et al., 2020; Al Ammari et al., 2018; Al-Hayani et al., 2021).

In Oman, a study investigating the genetic profile and drug resistance-conferring mutations in Mycobacterium tuberculosis found that expatriate status was a significant risk factor for drug-resistant TB, such as isoniazid resistance was lesser among the Omani nationals (p = 0.039, OR = 0.46; 95% CI [0.28–0.75]). Their results favor the hypothesis of migration as a possible source of resistant lineages in Oman (Al Mahrouqi et al., 2022).

Discussion

DR-TB represents an important challenge for global health, requiring comprehensive solutions in terms of diagnosis, treatment, and control strategies. Studies carried out in the GCC countries, such as KSA, Kuwait, Bahrain, Qatar, Oman, and UAE, shed some light on the distribution and determinants of drug-resistant TB in this region. Although variable rates of DR-TB have been observed in these countries, there are a lot of similar risk factors that are responsible for its development and dissemination. Resistance to second-line drugs, such as fluoroquinolones and injectable agents, poses a significant challenge in the management of MDR-TB, as it limits treatment options and complicates patient care. This pattern is not highly prevalent in GCC region. Nonetheless, it is critical for continuous monitoring to prevent further resistance (Hegazy et al., 2021; Varghese & Al-Hajoj, 2017; Habous et al., 2020).

In contrast to the GCC countries, other Asian countries like India, Pakistan, and Indonesia have to deal with huge problems related to DR-TB, most of which are facilitated by factors of high population density, poor healthcare infrastructure, and patient management challenges (Husain, Kupz & Kashyap, 2021; Lohiya et al., 2020; Khan et al., 2022; Sulistijawati et al., 2019). However, research carried out in the USA, and other Western countries is characterized by a different picture: lower prevalence rates in general but significant disparities among the vulnerable cross-sections, including immigrants, the homeless, and people living with HIV/AIDS (Gobaud et al., 2020; Mencarini et al., 2023). As compared to GCC countries, Egypt, Iran, and Iraq are other countries in the middle east and north Africa (MENA) region that have high TB burdens, including DR forms. Challenges in TB control are also due to the limited healthcare infrastructure, political instability, and population displacement. In such countries as Jordan, Lebanon, and Turkey, which host a huge number of refugees, extra barriers to TB control exist. Overcrowded refugee camps, poor health care access, and interrupted health services propagate TB transmission and prevent its diagnosis and treatment. The DR-TB disparities within and between countries in the MENA region are due to socioeconomic factors such as poverty, unemployment, and poor education. The groups that are most vulnerable, such as the refugees, the internally displaced people (IDPs), and the marginalized communities, are the ones who carry the largest burden of TB (Ahmad, Mokaddas & Al-Mutairi, 2018; Amin et al., 2024; Mohammed, Khudhair & Al-Rabeai, 2022).

In terms of prevalence, GCC countries have different epidemiological profiles that are affected by various factors, including population density, health infrastructure, and socioeconomic conditions. For example, in KSA, which is defined by a significant geographic and demographic diversity, regional differences in the rate of drug-resistant TB may be seen, which are connected with variations in access to health services and management practices (Al Ammari et al., 2018; Saati et al., 2021). Similarly, Qatar and UAE, with their high economic growth and urbanization, have unique TB control challenges, such as migrant populations and health system capacity. The risk factors related to DR-TB in the GCC countries are in line with the global trends and include such problems as improper treatment regimens, low patient adherence to medication, previous TB history, and socioeconomic disparities that include variations in expatriates’ population characteristics (Fantahun et al., 2023; Xi et al., 2022; Faustini, Hall & Perucci, 2006). Challenges in healthcare delivery often further worsen these risk factors, which include sometimes inadequate access to quality diagnostic and treatment services, particularly in remote or underserved areas.

The expatriate populations contribute to the considerable effects on the epidemiology of the DR-TB situation in the GCC countries and the larger Middle East and North Africa (MENA) region (Barry, 2021; Woldesemayat, 2021; Semilan et al., 2021). Moreover, the movement of the expatriate workforce within and across the GCC countries and the MENA region enables the spread of TB strains, including DR strains. Migration across borders and traveling for jobs increases the migration of TB strains between countries, making it more difficult to control the disease (Mowafi, 2011; Katoue et al., 2022). Policies for migrants’ health, focusing on TB screening, vaccination, and treatment for migrant populations, may be designed and put into practice. Cross-border cooperation between the GCC countries and neighboring countries is fundamental to controlling TB transmission among expatriate populations by building surveillance systems and sharing epidemiological data (Singh et al., 2022).

Another major challenge in the management of DR-TB, especially in GCC countries, with other regions being heavily burdened by the same, is inadequate medication adherence. In GCC countries, incomplete treatment is common for several reasons, including patient non-compliance with treatment regimens and discontinuation of access to healthcare. In our context, “incomplete treatment” refers to situations where patients do not complete the full prescribed course of antituberculosis medication, whether due to stopping treatment prematurely, missing doses, or discontinuing medication without medical guidance (World Health Organization (WHO), 2022b; Limenh et al., 2024). Conversely, low medication adherence meaning (not following prescribed treatment regimens) is common in the GCC countries (AlSahafi et al., 2019; Zhang, Gaafer & El Bayoumy, 2014). These are important problems for TB control efforts as they lead to the threat of treatment failure, relapses, and drug resistance. The resolution of incomplete treatment and low medication adherence needs intensive approaches aimed at eliminating the determining factors of non-adherence to treatment. These strategies include patient education, community-based outreach programs, social support services, and stigma reduction activities targeted at TB. Moreover, healthcare access, healthcare systems strengthening, and patient-provider communication should be addressed to enhance treatment adherence and reduce the burden of DR-TB. It is worth mentioning that the prevalence and complications associated with non-communicable diseases, such as diabetes, in GCC countries are increasing significantly. These pre-existing illnesses are important determinants of DR-TB identified in the present review (Sambas et al., 2020; Al Ammari et al., 2018; Habous et al., 2020). In addition to targeting vulnerable populations, additional screening measures can target people who have non-communicable diseases such as diabetes, which makes them more vulnerable to DR-TB.

In addition to the measures discussed earlier, better DR-TB control can be achieved in the whole GCC region when the healthcare infrastructure is invested in, including laboratory capacity, diagnostic tools, and trained healthcare personnel. Improvement of primary healthcare systems and provision of quality TB diagnosis and treatment services for all are critical components in health system strengthening efforts.

The present study utilized a standard method to find the prevalence and determinants of DR-TB in GCC countries. However, we acknowledge that our study has some limitations due to variability in diagnostic methods used in different studies. Lastly, limiting the review to English-language studies may have excluded relevant research published in other languages, introducing a language bias.

Conclusions

The present systematic review executed by the research team focuses on the ongoing problem of DR-TB in GCC countries, drawing attention to the prevalence rates and risk factors throughout the region. Medication non-compliance, immigrant factors, and socioeconomic discrepancies are all contributors that make DR a multidimensional problem. These determinants can be targeted through interventions such as advancements in diagnostic and treatment services for TB, infection control measures, and more investments in healthcare systems. Collaborative actions by relevant stakeholders are essential to implement evidence-based interventions that reduce the DR-TB burden and improve overall community health. Ongoing research and surveillance activities are necessary for monitoring DR patterns including, mono, multi-, and second-line DR, identifying new risk factors, and providing focused interventions to lessen the threat of DR-TB to public health in GCC countries.

Supplemental Information

Supplemental Information 1 PRISMA checklist.

Additional Information and Declarations

Competing Interests

Author Contributions

Data Availability

The authors declare that they have no competing interests.

Alaa Alibrahim conceived and designed the experiments, performed the experiments, analyzed the data, prepared figures and/or tables, authored or reviewed drafts of the article, PROSPERO REGISTRATION, and approved the final draft.

Homoud Alqahtani conceived and designed the experiments, performed the experiments, analyzed the data, prepared figures and/or tables, authored or reviewed drafts of the article, PROSPERO REGISTRATION, and approved the final draft.

Ashokkumar Thirunavukkarasu conceived and designed the experiments, performed the experiments, analyzed the data, prepared figures and/or tables, authored or reviewed drafts of the article, PROSPERO REGISTRATION, and approved the final draft.

Ibtisam Qazi performed the experiments, analyzed the data, authored or reviewed drafts of the article, and approved the final draft.

The following information was supplied regarding data availability:

This is a systematic review.

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
