# Peer review of "Prevalence, patterns, and determinants of drug-resistant tuberculosis in Gulf Cooperation Council countries: an updated systematic review"

_PeerJ, doi:10.7717/peerj.18628_

## Round 0.1 · original submission · Major Revisions

The authors should revise the manuscript based on reviewers' feedback to improve experimental design, result validity, grammar, and readability.

Reviewer 1 ·

Basic reporting

Although well written and understandable, it would be worthwhile to perform an English language revision to improve the quality of the article and eliminated unnecessary repetitions especially in the introduction section and discussion sections.
The abstract I suggest you write the first two sentences as follows (for clarity): “Drug resistance to antituberculosis drugs is a growing global problem that threatens the successful control of Tuberculosis (TB) both worldwide and within the Gulf Cooperation Council (GCC) countries. In the GCC, TB remains a significant public health issue. Understanding the prevalence and patterns of drug resistance to antituberculosis drugs is crucial for developing effective prevention and treatment strategies.”
Suggest the use of “drug resistant TB (DR TB)” instead of “TB drug resistant (TB DR)” and use one term constantly in the text.
Suggest to fine tune the last paragraph of the abstract to: “Isoniazid monoresistance was the most commonly found type of resistance, with varying degrees of prevalence of multidrug-resistant tuberculosis (MDR-TB). Risk factors for drug-resistant TB included diabetes mellitus, past TB treatment, younger age, female gender, and renal failure. There was a positive correlation between expatriate status and drug resistance. Collaborative actions by relevant stakeholders are essential to implement evidence-based interventions that reduce the TB-DR burden and improve overall community health.”
In the manuscript title correct “pattern” to “patterns”
Table 1 the legends need to be clearer – what does * mean, what does ** mean and what does “-“ mean?
In the risk factors section of the results, if possible could you please add the confidence intervals for those p-values if available?
I think the epidemiological data on TB burden could be based on recent reference of the global TB report of 2023, please update reference 2 and related data.

Experimental design

The research question is clearly identified in both the abstract and the introduction, emphasizing its relevance in the current context. The study aims to address the existing data gap on the evolution of drug resistance in TB within Gulf countries. Conducting surveillance studies would be ideal; however, these can be costly. As an alternative, a systematic review of existing data is a valid and less expensive methodology. Data collection was comprehensive, the inclusion criteria rigorous and the analysis was thorough to maximise reliability and applicability of findings. The research conformed to current ethical and systematic review standards.
I think the detailed search strategy for each database used should be made available as supplementary material to ensure reproducibility.

Validity of the findings

The studies on which the current findings were based on are fully provided including results. Conclusions made are supported by data and respond to the research question. However, the findings are based on studies with varying methodologies, and as such the strength and quality of findings is not as strong as wished for. However, the authors rightly acknowledged this.

Additional comments

No comment

Reviewer 2 ·

Basic reporting

As per basic reporting English language was written well. The lines 193-195 are bit confusing. I request the author to rewrite this sentence "Furthermore, renal failure was identified as a specific risk factor for rifampicin resistance, with a statistically significant association observed (p = 0.022)"

The background of the review, its search strategies, eligibility criteria of the articles are suitable for the title of the article. Inclusion of JBI critical appraisal checklist for considering the articles to write the review was the best attempt.

Experimental design

The study design and the consort diagram was well organised. Authors considering PRISMA guidelines is appreciable.

Validity of the findings

I would like to ask the authors if they have used any software to find p values for generating p-values.

Can this systemic review be considered as Meta-analysis

Additional comments

I request the author to address the following queries
1Q. The sentence in the lines 193-195 (Furthermore, renal failure was identified as a specific risk factor for rifampicin resistance, with a statistically significant association observed (p = 0.022) (4, 5, 16). ) seems to be confusing. Please rewrite the lines to make them meaningful sentence.
2Q. Line 259 carries “incomplete treatment”. Clearly define “incomplete treatment” in your terms.
3Q. The author has written Prevalence, determinants of Drug resistant TB well. Pattern of resistance might need few more inputs from Gulf Cooperation Council Countries. Why are resistance to second-line injectables and floroquinolones not reported in this review.
4Q. The results showed p value. How was it calculated. Whether the author has used any software for meta-analysis
5Q. What is meant by poly-resistance in the review

---

## Round 0.2 · accepted · Accept

In my opinion, and that of the reviewer, the authors have appropriately addressed the reviewers' concerns.

Reviewer 1 ·

Basic reporting

No further comment. Clear and professional English used

Experimental design

No further concerns.

Validity of the findings

No further comments

Additional comments

No comments